# A Simplified Finite Element Model for Seismic Analysis of a Single-Layer Reticulated Dome Coupled with an Air Duct System

Tianlong Zhang [image: ORCID] and Yuwei Zhou *

School of Civil Engineering and Architecture, Hainan University, Haikou 570228, China
* Correspondence: yuweizhou@hainanu.edu.cn

**Abstract:** Damage to nonstructural components, such as air ducts, in buildings during earthquakes, which are more fragile than single-layer reticulated domes, has a significant impact on the sustainability of the building's functionality. To study the coupling effect and failure mode of a single-layer reticulated dome with an air duct system, then, simplified finite element models of air ducts and flange bolt joints were established and validated against the solid element model. Moreover, the simplified finite element models of support hangers were also built and validated against the existing experiment. Three kinds of support hanger layout schemes were studied to analyze the dynamic characteristics and seismic responses of a single-layer reticulated dome with an air duct system from earthquakes at different intensities. The results showed that the simplified finite element model can effectively simulate the coupling effect and failure mode of the single-layer reticulated dome with an air duct system. The coupling effect of the air duct system reduces the natural vibration frequency in the dome and increases the number of damaged members in the dome by strong earthquakes. The rate of falling air ducts with all the seismic support hangers is the highest compared to the two other support hanger layout schemes.

**Keywords:** single-layer reticulated dome; air duct support hanger; seismic analysis; finite element model

## 1. Introduction

Nonstructural components are ancillary components in a building and can be divided into water pipes, air ducts, electrical equipment, etc. In the past, the seismic performance of the main structure was more highly considered in the research of long-span buildings. However, it was noted from much of the seismic damage data from around the world that the damage to the main structure of long-span buildings by earthquakes was relatively light, whereas the damage to the nonstructural components was more serious. For example, in the 2010 Chilean earthquake with a magnitude of 8.3, there was no significant damage to the main structure of Santiago Airport, yet its internal air conditioning equipment and firefighting pipes were severely damaged, causing the airport to stop operations and causing tens of millions of dollars in economic losses [1]. The seismic dynamic response by nonstructural components depends not only on the ground motion characteristics but also on the dynamic characteristics of the main structure. Long-span spatial structures have many degrees of freedom, dense distributions of natural vibration frequencies, and complex and coupled vibration modes. Moreover, vertical vibration is often the main mode of the first vibration. Considering the resonance and coupling effects between the main structure and nonstructural components, the dynamic response and failure of nonstructural components will be significantly amplified, resulting in severe earthquake losses and a reduction in overall sustainability. From many practical engineering cases, it is found that the single-layer reticulated dome structure often arranges more air pipes to increase air circulation in the dome's interior, and other pipelines, such as water supply and drainage

pipes are less arranged. As shown in Figure 1, air ducts are connected by flange bolt joints. They are located under the radial rods of a single-layer reticulated dome and are connected to the dome by support hangers.

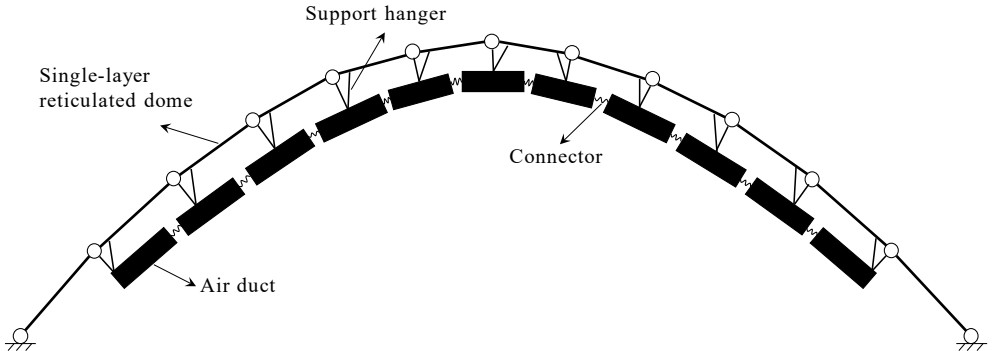

**Figure 1.** Single-layer reticulated dome with an air duct system.

An air duct system is a typical large nonstructural component, which has important functions, including ventilation, air supply, exhaust, and dust removal. Similar to other pipeline systems, there are three main types of damage to the air duct system by an earthquake, namely joint damage between pipelines, connection damage between pipelines and structures, and coupling damage between pipelines and other nonstructural components [2]. In the aspect of air pipe joints, some scholars have carried out related research on the mechanical characteristics and seismic performance of air pipe flange bolted joints. Kim et al. [3] studied the mechanical behavior of bolt flange joints under tensile load and believed that the maximum increase in bolt force was 40% of the preload. According to Wu et al. [4], the level of bolt preload and the friction coefficient of the interface have little influence on the axial tensile stiffness of the bolt flange, and the bolt preload mainly improves the resistance to flange separation. Couchaux et al. [5] studied the expressions of the initial rotational stiffness and the bending moment of a circular pipe connected by bolts and flanges under the bending moment and the axial action. Luan et al. [6] studied the force of a circular pipe connected with flange bolts under the action of a bending moment and proposed a simplified model composed of beam elements and spring elements. Chen et al. [7] designed a new flange bolt joint. Through testing, it was found that the joint was semi-rigid, capable of transferring part of the bending moment, and offered a good stiffness, bearing capacity, and energy dissipation capacity. Wang et al. [8] studied the mechanical properties of bolt flange joints under tensile, bending moment, and torsional loads, with the results showing that the more bolts, the larger the diameter, while the greater the pre-tightening force of the bolts, the greater the bending stiffness of the flange cylinder. However, the mechanical properties and hysteretic characteristics of flange bolt joints in air ducts have been less studied.

In another aspect of the connection between the air duct and the main structure, many scholars have carried out related research on the seismic support and suspension of air ducts. Goodwin et al. [9] conducted shaking table tests on hospital pipeline systems with or without seismic support hangers, to determine their deformation capacity and failure mode. The results showed that seismic support hangers reduced the displacement response by pipeline systems but could not reduce the acceleration response. Hoehler et al. [10] studied the seismic performance of seismic support hangers under different seismic excitations and found that the load of the load-bearing support hangers under the action of an earthquake was much greater than the anchorage force of seismic support hangers. To determine the seismic performance of pipeline systems with different forms of support, Tian [11] conducted dynamic tests on three groups of pipeline systems with different forms of support, and the results showed that in the dynamic tests with seismic support hangers, the degree of damage specimens with seismic support hangers was small. The suspension

screw, ceiling, spray joint, and pipeline joint of the pipeline system without seismic support and suspension were damaged. Wood et al. [12] studied the force–displacement hysteretic relationship of two types of commonly used longitudinal seismic support hangers and found that the performance of the support hangers under monotonic loading and cyclic loading had a great influence on their mechanical properties. Zhu et al. [13] proposed a coupling model of a structure–seismic support hanger and found that, compared to the equivalent lateral force method, the time–history analysis method can more accurately calculate the seismic action of the seismic support hanger in high-rise building structures. Qu et al. [14] analyzed the seismic response of the coupled system of a frame structure and large air duct equipment under the action of rare earthquakes; the results showed that in online elastic and elastoplastic analyses, the structure–equipment interaction would have a significant impact on the maximum response of the main structure. In summary, current research is on water pipelines and their coupling effects to the main structure by earthquakes; however, air ducts and their seismic coupling effects with other large-span space structures, such as single-layer reticulated domes were being less frequently studied.

As mentioned previously, this paper focuses on the coupling effect and failure mode of a single-layer reticulated dome coupled to an air duct system since the coupling effect of the air ducts is complex. In this study, the finite element (FE) software ABAQUS [15] was used to establish simplified FE models of the dome structure, air duct, and support hanger with the B31 elements and the model of the flange bolt joint with the connector elements. The simplified FE models of the flange bolt joints were validated against the solid element model, and the simplified FE models of the support hangers were validated against the existing experiment. Then, the seismic response and failure modes of the coupling FE models in three different support hanger layout cases were analyzed.

## 2. Establishment Method of the Simplified FE Model

### 2.1. General

The FE program ABAQUS [15] was used to establish a nonlinear FE model to simulate the behavior of a single-layer reticulated dome coupled with an air duct system. Flange bolts are mainly used to connect air ducts. The main damage to the air ducts by an earthquake is that the flange bolt cracks under the complicated action of a large shear force and a tension force, leading to the air ducts falling. To reduce the calculation time of the finite element model, the BEAM 31 element was used to simulate the air ducts in ABAQUS [15], which adopts the box section as the beam section, and the connector element was used to simulate the connection of the flange bolt joints in this paper, as shown in Figure 2.

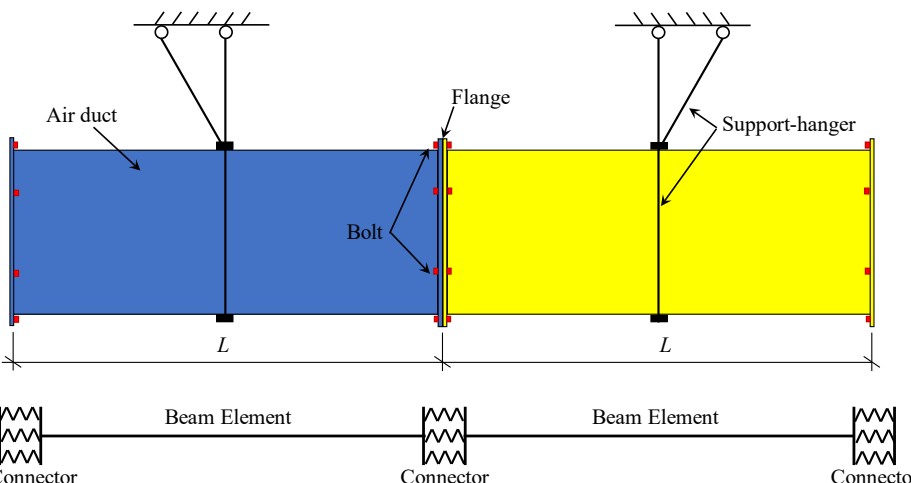

**Figure 2.** Simplified FE model of air ducts and flange bolt joints.

The connector element types selected were Cartesian and Align, that is, the connector has translational degrees of freedom in directions of $U_1$, $U_2$, and $U_3$, and limits the rotational degrees of freedom in directions of $UR_1$, $UR_2$, and $UR_3$. The connector element can define the complex mechanical relationship between two nodes. In this paper, the friction, damping, elasticity, and failure of the connector element are mainly defined according to the hysteretic performance of the flange bolt joint between the air ducts.

### 2.2. Simplified FE Model of Flange Bolt Joints

Using solid element C3D8R to simulate the mechanical behavior of bolted joints is a technique that has been adopted by many scholars [16–18]. C3D8R is used to establish a refined finite element model for the numerical analysis of the hysteresis properties of the air duct flange bolt joint by ABAQUS, as shown in Figure 3. The fine finite element model mainly includes two sides of the air ducts, flange, and bolts, where the unit length of the air pipe is 1 m, the section height of the air duct is 400 mm, the width of the air duct is 800 mm, and the duct thickness is 4 mm. The flange thickness is 4 mm, and the bolts are arranged as two bolts on each of the left and right sides and three bolts on the upper and lower sides. The bolt has a diameter of 8 mm and a length of 33 mm. The numbering and arrangement of the bolts are shown in Figure 3b.

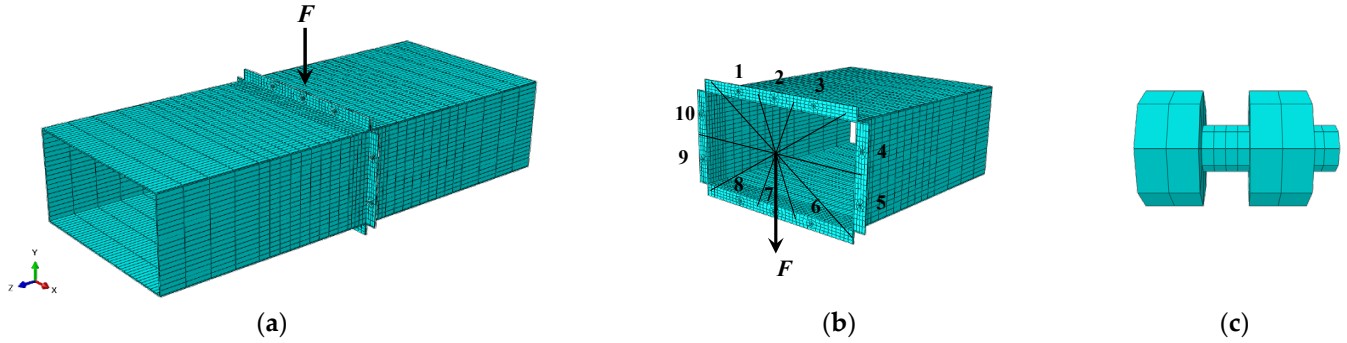

| (**a**) | (**b**) | (**c**) |

**Figure 3.** Solid finite element model of rectangular air duct flange bolt joint: (**a**) overall model; (**b**) air duct flange model; (**c**) bolt model.

The coupling point is located at the center of the flange face of one side of the duct. A vertical shear *F* is added at the coupling point to analyze the hysteretic curve and skeleton curve in the bolt group under seismic shear. The two ends of the model are coupled to the central reference points, respectively, and the boundary conditions of the coupling points are set as symmetric constraints to simulate the limiting effect of the seismic support hangers in the displacement of the duct in each direction. The contact in ABAQUS is arranged on the surface between the bolt and flange. There are two analysis steps in the static analysis. The first step applies gravity and bolt preload, and the second step applies cyclic shear load *F*. To improve the computational efficiency and accuracy of the simulation, the mesh of the flange and bolt contact parts was finely divided. Furthermore, the mesh density of the duct was gradually reduced from one end of the flange to the other end.

The air duct is made of Q235 steel, with an elastic modulus of $2.1 \times 10^5$ MPa, a Poisson ratio of 0.3, and a mass density of 7850 kg/m³ [19]. The elastic–plastic metal constitutive model is adopted for the duct material, and the specific material parameters are shown in Table 1.

**Table 1.** Stress–strain parameters of the air duct material [20].

| Yield Stress | Plastic Strain |
|---|---|
| 235 MPa | 0 |
| 311 MPa | 0.007325 |
| 353 MPa | 0.015873 |
| 377 MPa | 0.024435 |

The bolts are made of 40 Cr alloy steel with an elastic modulus of $2.06 \times 10^5$ MPa, a Poisson ratio of 0.29, and a mass density of 7820 kg/m$^3$. The elastic–plastic metal constitutive model was also adopted for the bolts, and the flexible damage and damage evolution were set. The damage evolution takes the form of displacement, and the failure displacement is 1 mm. The specific elastic–plastic parameters of the bolts are shown in Table 2.

**Table 2.** Stress–strain parameters of the bolt material [21].

| Yield Stress | Plastic Strain |
|:---:|:---:|
| 785 MPa | 0 |
| 935 MPa | 0.001 |

To test the precision of the constitutive model for the bolt material, numerical simulations of tensile and shear failure for the individual bolts were carried out using ABAQUS. The numerical simulation was carried out with reference to the experiments of Guzas et al. [22]. Figure 4 shows the fracture pattern and the force–displacement curve for the tensile failure of a single bolt. It can be seen from Figure 4 that the failure location is concentrated in the bolt screw, which exhibits significant shrinkage during the tensile process. The maximum tensile capacity of a single bolt is 55 kN, and the bolt screw is eventually broken by the tensile force. In general, the numerical simulation results were close to the experimental results. Therefore, the selected bolt material constitutive model can effectively simulate the peak value and decreasing process of bolt tensile bearing capacity.

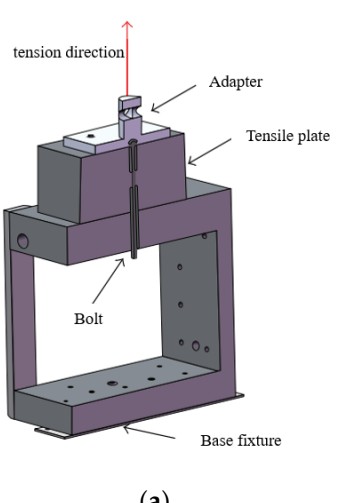

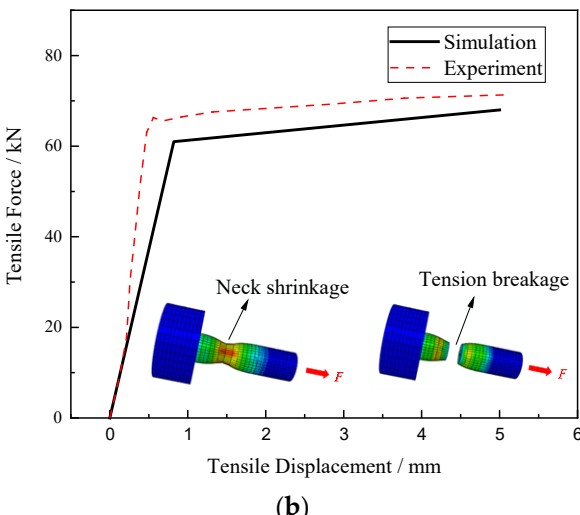

(**a**)  (**b**)

**Figure 4.** Numerical simulation of tensile failure by a single bolt: (**a**) static tension test; (**b**) force–displacement curves.

In this study, the numerical simulation of the bolt double shear test was carried out, in reference to Guo et al. [23]. The shear failure mode and its force–displacement curve for a single bolt under the action of two steel flanges are shown in Figure 5. It can be seen from Figure 5 that the failure location was in the single shear plane of the bolt crew and the maximum shear bearing capacity of the single bolt was 30 kN. Compared to the experimental result by Guo et al., the curve obtained in this paper is similar to the experimental one, which also shows the accuracy of the constitutive model of the bolt material. The numerical simulation results also showed that the end plates were partially damaged when the holes in the end plates were being compressed, which indicated that the damage to the flange plates around the bolts must be considered when the bolt is completely cut.

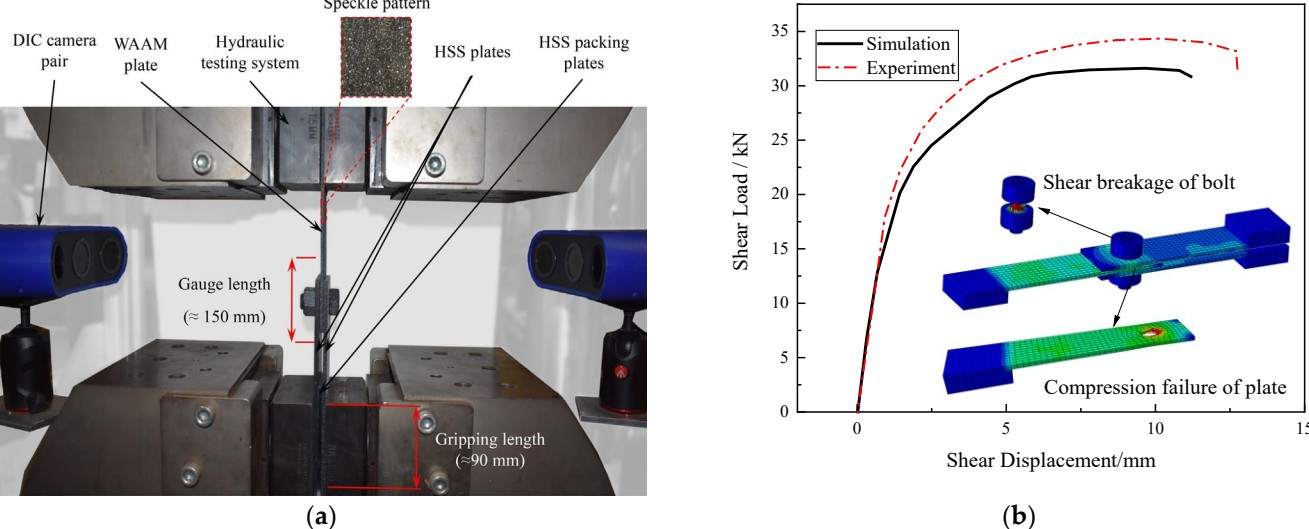

(**a**)                                                          (**b**)

**Figure 5.** Numerical simulation of shear failure by a single bolt: (**a**) static shear test; (**b**) force–displacement curves.

The numerical simulation of the bolt failure above verifies the accuracy of the material structure of the bolt. Therefore, the quasi-static numerical simulation was further used to carry out the cyclic loading of the fine finite element model of the rectangular air duct flange joint. The loading curve is shown in Figure 6a, which shows that the time step is 40 s and cyclic displacement gradually increases from 0 to 10 mm. In addition, a preload of 25 kN was applied to each bolt to ensure the bolt was connected to the air duct. The specific loading system and hysteresis curve are shown in Figure 6. It can be seen from the hysteresis curve in Figure 6b that the hysteresis curve of the flange bolt joint in the air duct was an inverted S-shape, which means it has a strong rheostriction effect and a long slip segment. The results showed that the joint was affected by more bolt slippage under vertical shear force, and the ductility and energy dissipation capacity were weak.

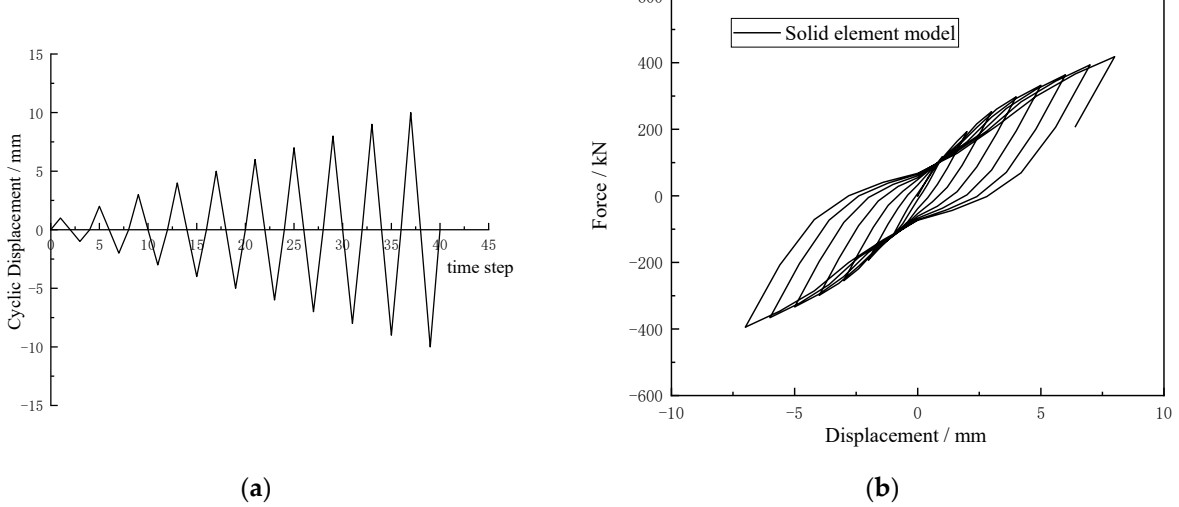

(**a**)                                                          (**b**)

**Figure 6.** Numerical simulation of a single air duct: (**a**) loading curve; (**b**) hysteretic curve.

Through the hysteretic simulation by ABAQUS, the final failure mode of the flange bolt joint under the action of the cyclic shear force is shown in Figure 7, where the loading displacement is 7 mm. According to the analysis in Figure 7, the stress on the air duct and the flange is lower, and the stress on the bolt is higher. The damage occurred mainly in the bolts. The bolts on the left and right sides of the flange had large shear deformation and

tensile deformation, and the failure occurred under the combined action of the tension and shear force. The bolts on the upper and lower sides of the flange exhibited a certain degree of tensile deformation in the hysteresis simulation, without tensile failure or shear failure, indicating that the stress was relatively lower than those on both sides.

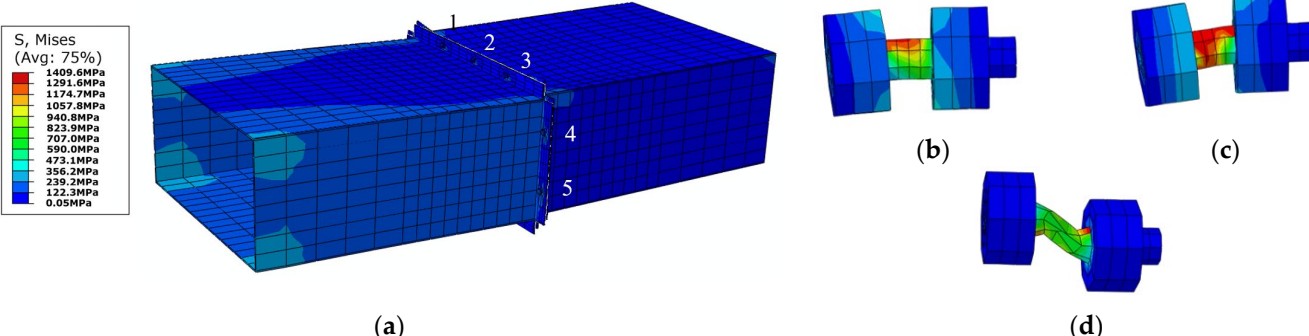

(a)

(b)

(c)

(d)

**Figure 7.** Hysteresis simulation result of the air duct flange bolt joint: (**a**) stress distribution; (**b**) failure mode of bolt no. 2; (**c**) failure mode of bolt no. 3; (**d**) failure mode of bolt no. 4.

According to the hysteresis simulation results for the air duct joint, it is feasible to simplify the air duct because the force of the air duct is much lower than the bolts and no damage occurs in the ducts. At the same time, due to the failure of the bolts, the connector element was used to simulate the tensile shear failure of the air duct flange bolt joint. The connector element is a versatile connection element that can simulate a wide range of connection behaviors in ABAQUS, including elasticity, plasticity, and damage. Additionally, the spring element is one of the connection elements in ABAQUS. The connector element can define a variety of behavioral attributes through the connector section. However, we can only define the linear or nonlinear stiffness of the spring in the spring element. Therefore, we chose the connector element to model the mechanical behavior of the flange bolt joints in this study. The connection type of the connector element was Cartesian, which means that the connector provides a connection between the two air duct nodes, which allows for independent behavior in each of the three local Cartesian directions that follow the system at the node of one side of the air duct. In the connector section, we defined the friction, damping, elasticity, and failure of the connector element. The elastic modulus in each direction of the connector element was derived from the initial slope of the force–displacement curves obtained by the tension simulation and shear simulation for a single bolt. By defining the axial elastic–plastic behavior of the connector element in the axial direction, the force–displacement curve of the bolt was simulated. By defining the nonlinear elastic behavior of the connector element in the shear direction, the shear slip effect of the bolt was simulated. The fracture failure of the bolt was simulated by defining the tensile limit displacement $U_{max}$ of the connector element. The simplified mechanical model of the air duct flange bolt joint in this paper is shown in Figure 8a, and the comparison of the hysteresis curves obtained from the two models by using the connector element and solid C3D8R element is shown in Figure 8b. As shown in Figure 8a, the elasticity of the connector in the tangential direction was set to nonlinear in the connector section manager, and the deformation to its stiffness decreased when the displacement reached the plastic onset, thereby simulating the mechanical behavior of the flange bolt joint in the tangential direction. The results showed that the hysteresis curve for the simplified model had a more serious rheostriction phenomenon, which is a conservative calculation compared to the solid element. When the vertical displacement was loaded to 7 mm, the bearing reaction force on the left side of the simplified model was 375 kN, while that of the solid element model was 397 kN. The value of the simplified model was about 5.5% lower than for the solid element model, and the error was very small. With the same hardware and software, the solid element model took 34 min to compute, and the connector element model took

21 min, which showed the higher computational efficiency of the connector element model. In summary, the simplified model can accurately simulate the hysteresis characteristic of the flange bolt joint in the air duct and is faster than the solid element model in calculation, meaning that it can be applied in subsequent research.

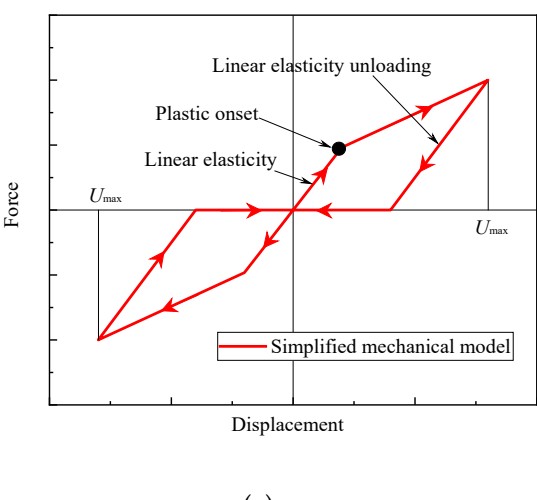

(**a**)

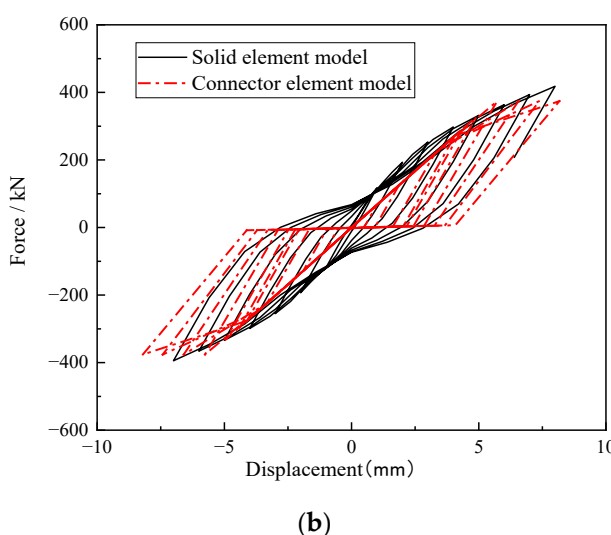

(**b**)

**Figure 8.** Force–displacement curves of the air duct flange bolt joint: (**a**) simplified mechanical model; (**b**) connector element model.

### 2.3. Simplified FE Model of Support Hangers

Support hangers are used in the buildings to bear the self-weight of nonstructural components, such as pipes and ducts, and are firmly connected to the main structure. Support hangers are generally divided into ordinary support hangers and seismic support hangers. Compared to the ordinary support hangers, the seismic support hangers add seismic bracings in two horizontal directions, which can effectively bear the horizontal forces and limit the horizontal deformation of the pipes and air ducts under earthquake conditions, control the vibration of nonstructural components, and finally, protect the nonstructural components. The seismic support hangers are equipped with two longitudinal bracings and one lateral bracing with the same section and the same angle (45°). Furthermore, the only difference between the longitudinal bracings and the lateral bracings is their restriction of the direction of the air duct movement. In this paper, the BEAM 31 element was used to conduct finite element modeling of ordinary support hangers and seismic support hangers, and the established finite element models are shown in Figure 9. The bracings, vertical hanging rods, and horizontal clamping bars are all channel-section members with a height and width of 41 mm and a thickness of 2 mm, while the vertical clamping bars adopt a circular section with a radius of 6 mm to fix the left and right sides of the air duct. The connections between each member of the support hanger system and the connections between the support hanger and the main structure are all hinged. To simulate the buckling failure process, the beam elements of each member were meshed into four elements. Tie constraints were used to simulate the connection between the air ducts and the support hanger members because the air ducts are firmly bound to the support hangers by bolts.

To verify the accuracy of the simplified model of the seismic support hangers, a numerical simulation of the static experiment, in reference [24], was carried out using the simplified model. The material of the bracket members was Q235 steel. Hinged constraints were established at the end of the members and brace, and the same gradually increasing horizontal force was established at both ends of the horizontal channel steel. The skeleton curve obtained from the test and the force–displacement curve simulated by the simplified model in this paper are shown in Figure 10b. In the stress figure in Figure 10b, a redder color indicates higher stress and a bluer color indicates lower stress. The test results reported by

Song et al. [23] suggested that when the lateral displacement of the loading point is 50 mm, the test load is 13.05 kN, while the numerical simulation load is 14.2 kN, which is 8.8% larger than the test value. When the test load is loaded to 17.7 kN, the brace buckling fails, while the numerical simulation failure load is 7.74 kN, which is only 0.2% higher than the test value. Therefore, the simplified finite element model of the seismic support hangers established in this paper can effectively simulate the actual stress of the seismic support hangers under seismic horizontal shear and can be used in future research.

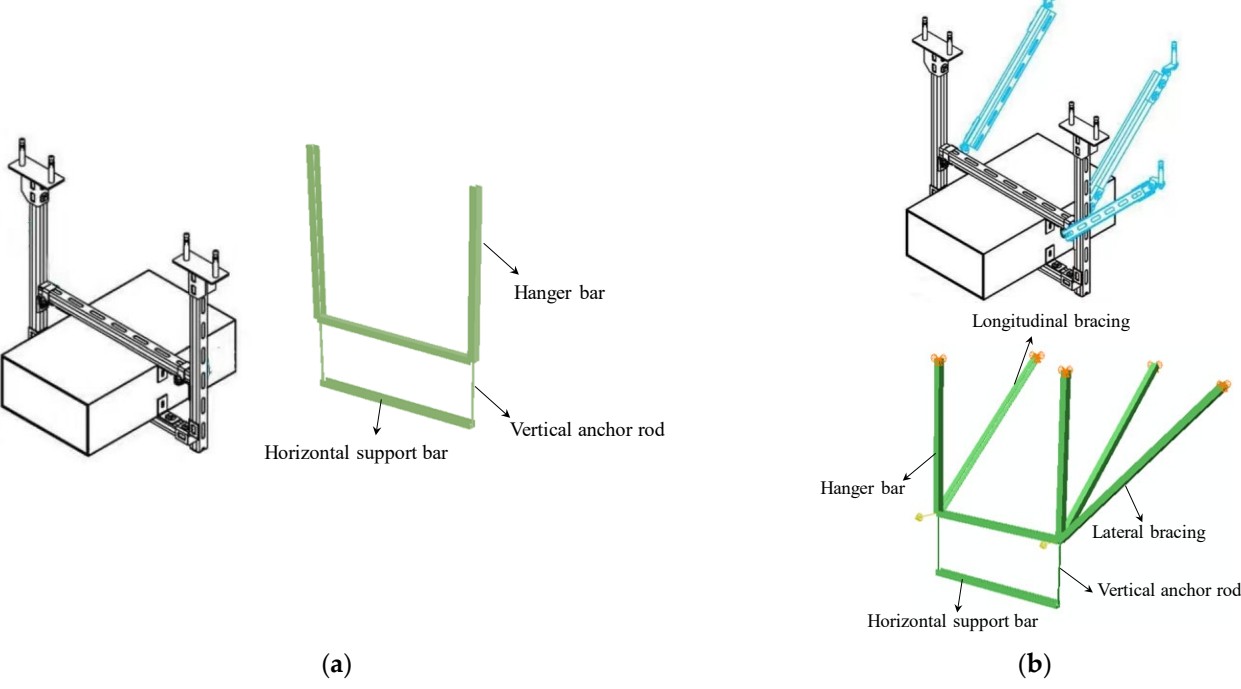

(**a**)　　　　　　　　　　　(**b**)

**Figure 9.** Simplified finite element model of duct support hangers: (**a**) ordinary support hangers without bracings; (**b**) seismic support hangers.

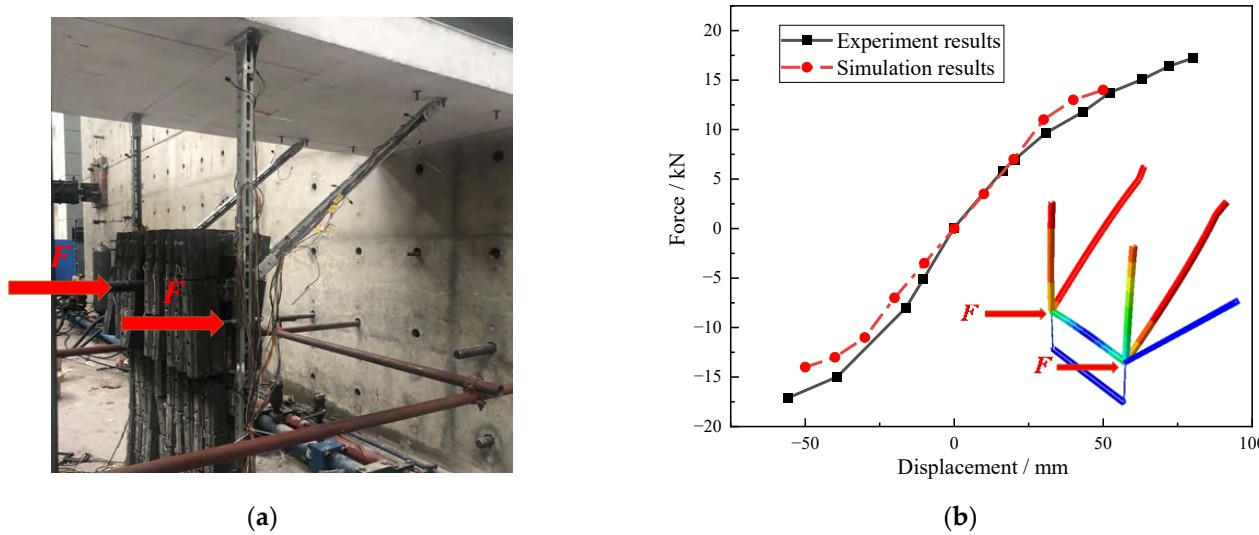

(**a**)　　　　　　　　　　　(**b**)

**Figure 10.** Comparison of experiment and numerical simulation: (**a**) experiment photograph; (**b**) force-displacement curves.

## 3. Case Study and Results

### 3.1. FE Model of the Single-Layer Reticulated Dome

The Kiewitt-8 single-layer spherical reticulated dome was selected as the research object in this paper, as shown in Figure 11. The span of this dome is 40 m, and the rise-to-span ratio is 1/4. The boundary conditions include fixed hinge supports, and the dome material is Q235 steel, with a yield strength of 235 MPa, a density of 7850 kg/m$^3$, and an elastic modulus of $2.06 \times 10^5$ MPa. There are six annular circles and the dead load of the roof is 1.0 kN/m$^2$. Different pipe sections of the single-layer reticulated dome are shown in Table 3. The finite element software ABAQUS was also used to establish the dome model. The bars in the dome were rigidly connected and the mass element was set at each joint. The element type of the dome bar adopted the Beam 31 element, and each bar was meshed into six elements. The constitutive model of the steel material is the ductile metal elastic–plastic damage model, and the constitutive parameters were derived from reference [18].

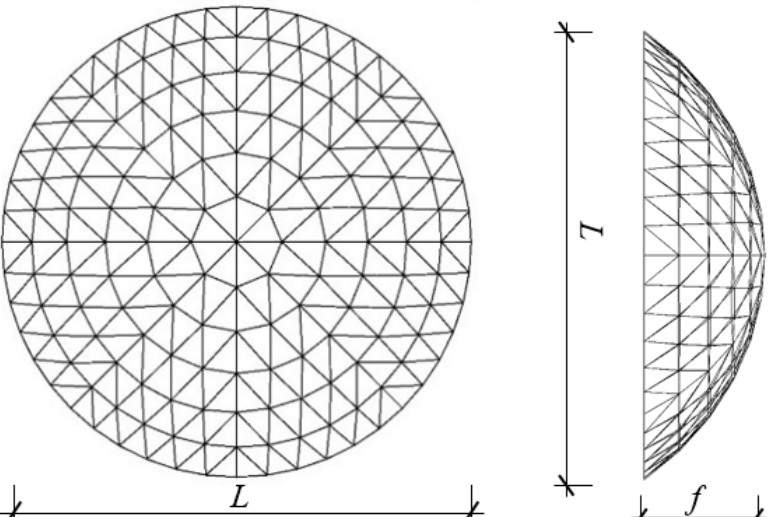

**Figure 11.** Case model of the single-layer spherical reticulated dome.

**Table 3.** Pipe sections of the single-layer reticulated dome.

| Dome Member Type | Pipe Section |
| --- | --- |
| Radial bar | Φ 133 × 4 |
| Annual bar | Φ 133 × 4 |
| Diagonal bar | Φ 114 × 3 |

### 3.2. FE Model of Air Duct System

According to the practical case of the Qionghai City Stadium in China, the specific arrangement of the air duct system is shown in Figure 12. The air duct system has a total of eight air ducts, which are arranged under the radial bars in the dome. Each air duct is divided into two parts: the main air duct is arranged in the radial direction along the dome, and the branch air duct is arranged in the annual direction along the dome. The bottom of the main air duct is the air inlet, and the end of the branch air duct is the air outlet. The total length of each main air duct is 14.9 m, and the length of each branch air duct is 3.1 m. The rectangular section parameters of the main and branch air ducts are consistent with those in Section 2.2.

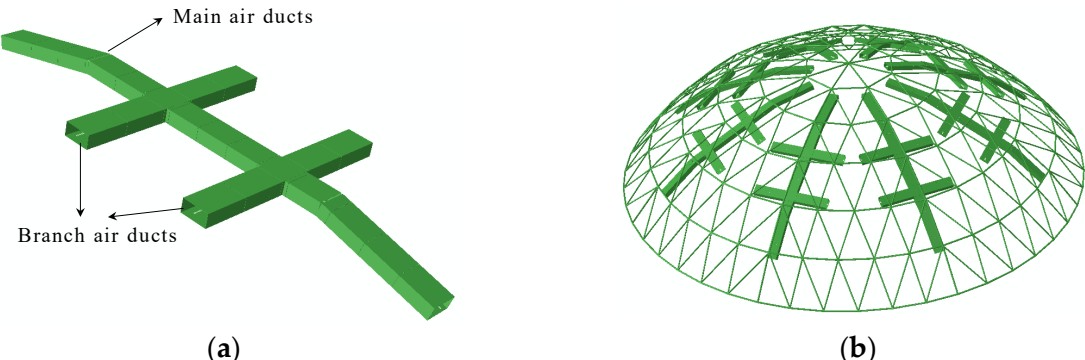

**Figure 12.** Simplified FE model of air ducts: (**a**) single air duct model; (**b**) all air duct models.

To inspect the influence of the different bracket layouts on the support hangers' response to the dome structure and air ducts, three types of bracket layout schemes were selected as the research objects in this paper, as shown in Figure 13. Among them, is case 1, where all the brackets of the main air duct are ordinary brackets without bracings. Case 2, where all the brackets of the main air duct are seismic support hangers with bracings. Case 3, where the staggered arrangements are used with the ordinary brackets and the seismic support hangers. In case 3, the distance between the seismic support hangers is 8.1 m, which meets Table 8.2.3 in the Chinese Code GB 50981-2014 [25], whereby the maximum distance between lateral seismic support hangers for air ducts of ordinary rigid materials in new construction projects is 9 m, and the maximum distance in the longitudinal direction is 18 m. Due to the short length of the branch air duct, ordinary brackets are used on them. Finally, the simplified finite element model of the reticulated dome, air ducts, and brackets is assembled together. The connector element established in this paper was used to simulate the flange bolt joint between the air ducts. The connection between the air ducts and brackets is bound by the tie constraint in ABAUQS and it is assumed that the rectangular air duct can be restrained by the right, left, up, and downsides of the bracket members and bolts. The connection between the brackets and the dome is hinged. In accordance with Section 8.2.2 in the Chinese code GB 50011-2010 [26], the damping ratio in the elastic time–history analysis of the whole FE model is 0.02, and that in the elastic-plastic analysis is 0.05, which is input into the material manager in ABAQUS.

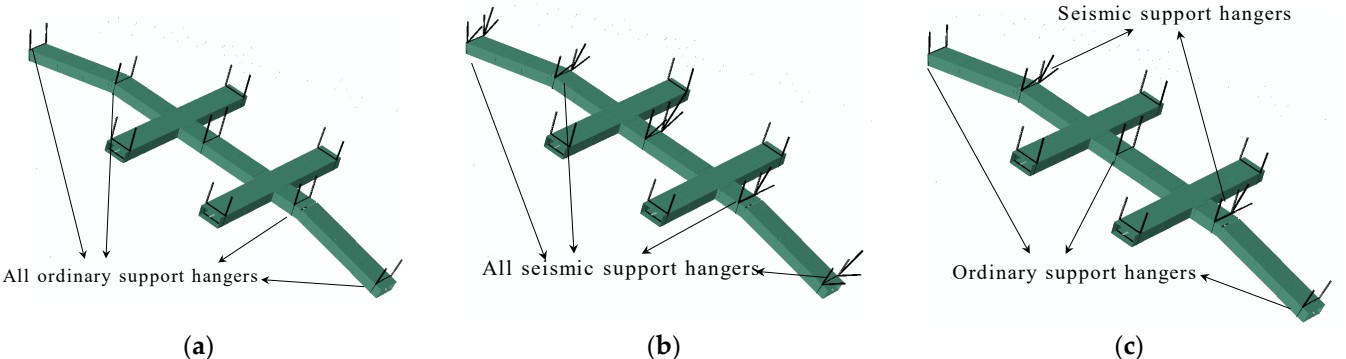

**Figure 13.** Support hanger layout schemes: (**a**) case 1: ordinary support hangers without bracings; (**b**) case 2: seismic support hangers with bracings; (**c**) case 3: staggered arrangement with the ordinary support hangers and the seismic support hangers.

### 3.3. Results of Dynamic Characteristics Analysis

Modal analysis is a common method applied to studying the dynamic characteristics of a structure. Modes are the inherent vibration characteristics of a structure, and each mode has a specific natural vibration frequency and deformation. In this paper, the mode analysis of a single-layer reticulated dome without an air duct system, which is called

case 0, and the dome coupled with an air duct system in three support hanger layouts, which are called case 1, case 2, and case 3, were carried out, and the first three modes and frequencies from the four cases were acquired, as shown in Table 4. The results in Table 4 show that the natural vibration frequency of case 1 was the highest in the first three modes, which means that the stiffness of the single-layer reticulated dome without any ducts was the maximum and the overall stiffness of the single-layer reticulated dome coupled with an air duct system will become smaller. The results also show that by comparing the three cases of the coupling air duct systems, the natural vibration frequency of case 1 was the lowest, while that of case 2 was the highest, and for case 3 it was located between case 1 and case 2. This illustrates that the overall stiffness of a dome fully arranged with ordinary support hangers without bracings is lower, and the overall stiffness of a dome fully arranged with seismic support hangers with bracings is higher. The overall stiffness of a dome arranged with the ordinary support hangers and the seismic support hangers is located between the two preceding cases.

**Table 4.** The natural vibration frequency for the four cases.

| Dome Case | First Mode Frequency | Second Mode Frequency | Third Mode Frequency |
|---|---|---|---|
| Case 0 | 4.1854 | 4.1854 | 5.0724 |
| Case 1 | 3.5365 | 3.6262 | 3.7547 |
| Case 2 | 3.8531 | 3.8591 | 5.0135 |
| Case 3 | 3.8088 | 3.8322 | 4.6323 |

### 3.4. Results of the Seismic Time–History Response Analysis

According to the Chinese code GB 55002-2021, Table 2.1.2 [27], seismic hazards are defined as frequent earthquakes, moderate earthquakes, and rare earthquakes. Moreover, the probability of frequent earthquakes occurring is 63% beyond the 50-year probability, the probability of moderate earthquakes occurring is 10% beyond the 50-year probability, and the probability of rare earthquakes occurring is 3% beyond the 50-year probability. Taking the peak ground acceleration (PGA) as the index of ground motion intensity, 3D earthquakes, including the El Centro Earthquake wave, Taft Earthquake wave, Loma Prieta Earthquake wave, and Tianjin Earthquake wave, were adopted to conduct an elastic–plastic time–history analysis. The PGA of the seismic waves was adjusted to 110 cm/s$^2$, 300 cm/s$^2$, and 510 cm/s$^2$. The PGA ratio in the X, Y, and Z directions was 1:0.85:0.65. ABAQUS was used to simulate the dynamic response of the four cases, and the maximum displacements of the dome nodes in the three directions were selected as the dynamic response index.

The maximum vertical displacements of the dome nodes in four cases under different earthquake waves are shown in Figure 14. The results in Figure 14 show that the maximum vertical displacements of the four cases were different. When the PGA was 110 gal, the maximum vertical displacements of the four cases were 6.42 mm, 11.13 mm, 5.88 mm, and 6.38 mm, respectively, which illustrates that case 1 increased by 73.36% compared to case 0, while the data for case 2 and case 3 hardly increased. When the PGA was 300 gal, the maximum vertical displacements of the four cases were 17.91 mm, 20.09 mm, 16.83 mm, and 18.39 mm, respectively, which indicates that case 1 increased by 12.17% compared to case 0, whereas case 2 and case 3 hardly increased, thereby showing the same pattern as above. When the PGA was 510 gal, the maximum vertical displacements of the four cases were 30.97 mm, 31.60 mm, 30.08 mm, and 32.45 mm, respectively, which means the air duct system has little influence on the vertical displacements in the dome structure under rare earthquakes. This is because the vertical stiffness in the air duct system is very small compared to the dome structure, meaning the air duct system will be destroyed first. To sum up, considering the coupling effect in the air duct system, the layouts of all ordinary support hangers have a great impact on the vertical displacement of the dome structure under frequent earthquakes, a certain impact under medium earthquakes, and little impact

under rare earthquakes. However, there is little effect on the vertical displacement of the dome structure under frequent earthquakes, moderate earthquakes, or rare earthquakes when the seismic support hangers are adopted in whole or in part.

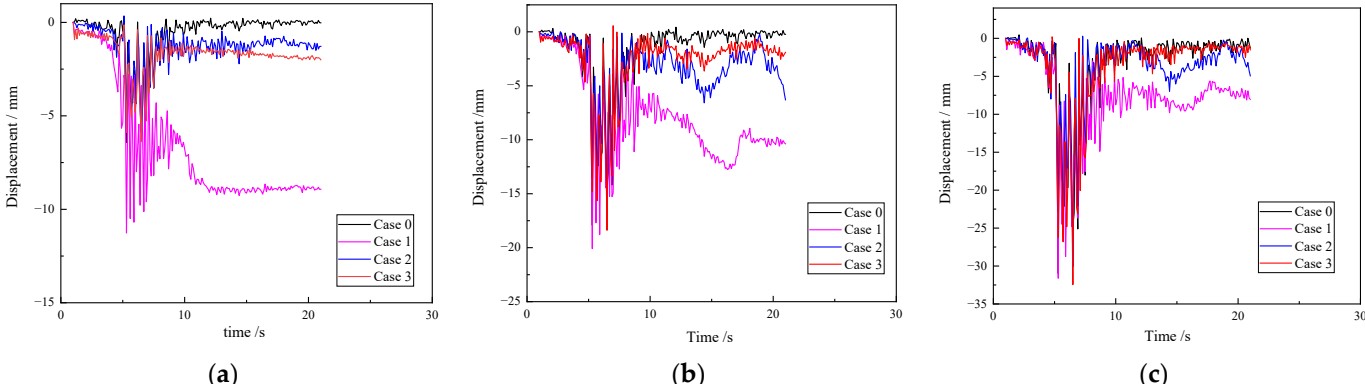

(**a**)  (**b**)  (**c**)

**Figure 14.** Maximum vertical displacement curves of the dome nodes: (**a**) frequent earthquake (PGA = 110 gal); (**b**) moderate earthquake (PGA = 300 gal); (**c**) rare earthquakes (PGA = 510 gal).

Since only a small amount of the support hangers in the four cases showed plastic deformation when the PGA was equal to 510 gal, and neither the dome members nor the air ducts were damaged, the IDA time–history analysis method was adopted in this paper to increase the amplitude of the PGA of the earthquake waves to 1300 gal, and investigate the failure mode of the coupled system of the single-layer reticulated dome, the air ducts, and the support hangers under strong earthquake excitations. The positions of the damaged members of the dome and the falling distribution modes of the air ducts are shown in Figure 15. By calculating the number of buckling members in each dome case, the number of dome buckling members in case 0, case 1, case 2, and case 3 were obtained as 0, 7, 3, and 2, respectively. The results showed that the dome without a coupled air duct system did not cause any damage to the members under strong earthquakes and the number of damaged members in the dome with the coupled air ducts of all ordinary support hangers was the largest. The damaged bars in case 1 were mainly distributed in the outermost second annual bars and the diagonal bars connected with the fixed hinged support. The damaged bars in cases 2 and 3 were mainly distributed in the diagonal bars connected to the support. Therefore, considering that the coupling effect of the air ducts will increase the number of damaged members in the dome under strong earthquakes, different support hanger layouts will also affect the number of members in the dome.

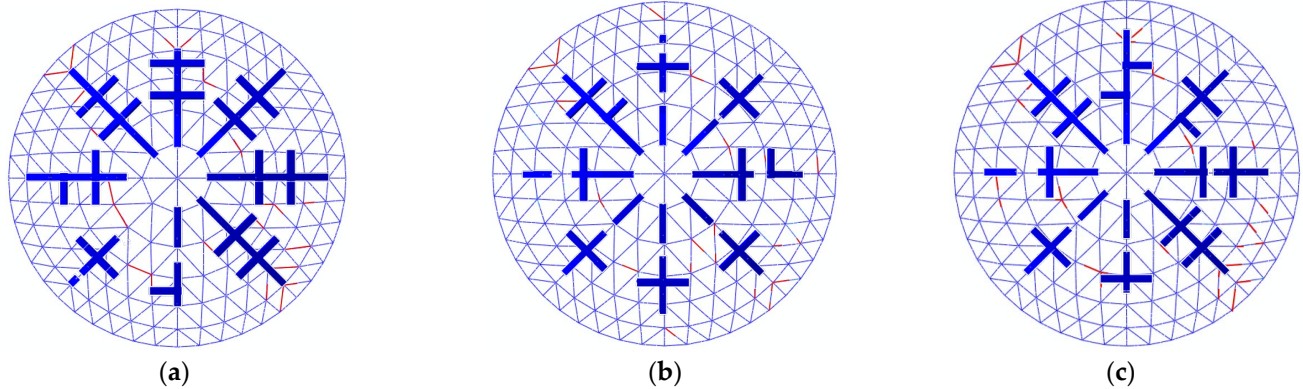

(**a**)  (**b**)  (**c**)

**Figure 15.** Falling distributions of air ducts in different cases: (**a**) case 1; (**b**) case 2; (**c**) case 3.

By defining the ratio of the length of the falling ducts to the total length of the whole air duct as the falling rate, the falling rate of air ducts in case 1, case 2, and case 3 were, 13.6%,

25.6%, and 17%, respectively. The results showed that the ducts would be destroyed before the dome structure under a strong earthquake. For the dome with only ordinary support hangers, the falling rate of the air ducts under a strong earthquake is the lowest and for the dome with only seismic support hangers, the falling rate of the air ducts under a strong earthquake is the highest. This is because the horizontal stiffness of the seismic support hangers is larger than that of the ordinary support hangers, the greater the horizontal forces transmitted by the support hangers to the air ducts under earthquake actions, the more the air ducts fall. Therefore, the falling rate of the air ducts under the action of a strong earthquake also has a great relationship with the different layout schemes of the support hangers.

## 4. Conclusions

In this study, the simplified finite element models of the air ducts and flange bolt joints were established and verified by the fine solid element model. Three kinds of support hanger arrangement schemes were considered and the coupling models for a single-layer reticulated dome with air duct systems were also established. The natural vibration frequency analysis and seismic time–history analysis of four cases were carried out and the damage mode of the dome and the falling rate of the air ducts were analyzed. The main conclusions are summarized as follows:

(1) The hysteresis curve of the flange bolt joint in the air ducts simulated by a fine solid element was S-shaped, with a strong pinching effect and a relatively long sliding segment, and the energy dissipation capacity and ductility of the joint were relatively weak. The simplified mechanical model of the flange bolt joint established by the connector element can effectively simulate the pinching effect and hysteresis curve of the joint and is faster than the solid element model in numerical calculation.

(2) The air duct system has an obvious influence on the dynamic characteristics and natural vibration frequency of the single-layer reticulated dome. Compared to the single-layer reticulated dome model without an air duct system, the maximum vertical displacement of the dome nodes with all ordinary air duct support hangers increased by 73.36% under frequent earthquakes, 12.17% under moderate earthquakes, and 2.03% under rare earthquakes. The maximum vertical displacement of the dome nodes was less affected by the air duct system with full or partial seismic support hangers. The results illustrate that the full or partial use of seismic support hangers can preferentially reduce the vertical displacement of air ducts. Therefore, the coupling effect of nonstructural components, such as air ducts and support hangers, on the main structure should be considered in response to a single-layer reticulated dome under earthquake excitation.

(3) Under strong earthquakes, when the PGA was equal to 1300 gal, the dome model without the air duct system caused no damage to any member, yet considering the coupling effect of the air duct system, some members in the dome models in three cases were damaged and some air ducts fell, indicating that the coupling effect of the air duct system will cause damage to the dome model in advance and affect the sustainability of the building. Among the three cases of the hanger arrangement, the falling rate of the air ducts was the highest with all the seismic hangers when the dome model was under strong earthquake, which shows that a single-layer reticulated dome with staggered seismic conditions and an ordinary support hanger is the best hanger arrangement due to the minimal damage to the dome and the lower falling rate of the air ducts. For this reason, when arranging seismic hangers, it is necessary to consider the characteristics of the air ducts to improve the sustainability of the dome and air duct system.

**Author Contributions:** T.Z.: Conceptualization, methodology, supervision, and writing—original draft preparation; Y.Z.: Software, validation, and writing—review and editing. All authors have read and agreed to the published version of the manuscript.

**Funding:** This research was funded by the Hainan Provincial Natural Science Foundation of China (Grant No. 520QN232).

**Institutional Review Board Statement:** Not applicable.

**Informed Consent Statement:** Not applicable.

**Data Availability Statement:** Not applicable.

**Conflicts of Interest:** The authors declare that they have no known competing financial interests or personal relationships that could have appeared to influence the work reported in this paper.

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
