# Peer review of "A Simplified Finite Element Model for Seismic Analysis of a Single-Layer Reticulated Dome Coupled with an Air Duct System"

_sustainability, doi:10.3390/su151712902_

Round 1

Reviewer 1 Report

This is a carefully done study and the findings are of considerable interest.

The paper emphasized the influence of the air duct system on a single-layer reticulated dome under earthquake excitations. ABAQUS finite-element programme was used to established the simplified model and verify the analysis result. Some minor revisions are list below.

(1)  The type and the property definition of the Connector element in Section 2.2 are not clearly descripted by authors. And authors should explain why you chose the Connector element and what the difference between the Connector element and the Spring element.

(2)  The boundary condition and the loading method in Figure 3 are not mentioned in Section 2.2.

(3)  The hysteresis curves in Figure 8(b) are unclear between two models. This figure must be redrawn to let the readers distinguish the curves inside. And authors should explain the computational efficiency and accuracy between these two models.

(4)  What is the difference between the lateral bracing and the longitudinal bracing in Figure 9? However, the lateral bracing is only mentioned in Section 3. The missing description of the longitudinal bracing must be completed in Section 3.

(5)  Other major drawbacks of the paper are given in the following. The section geometry of the seismic support hangers in Figure 10 is completely missing and must be included.

(6)  The damping ratio of 0.2 is not accurate enough for elastic-plastic analysis in Line 315.

(7)  All the codes in Section 3.2 and Section 3.3 in the article are Chinese codes, but the reviewer and possibly most of the readers will not be familiar with Chinese standards. The author must describe the specific terms of these mentioned codes.

(8)  English language and style are minor spell check required. For example, the title of Section 2.1 should be called “General”. And the naming of the single-layer reticulated dome and seismic support hangers should be respectively consistent in this paper.

(9)  The format of reference is inconsistent.

This is a carefully done study and the findings are of considerable interest.

The paper emphasized the influence of the air duct system on a single-layer reticulated dome under earthquake excitations. ABAQUS finite-element programme was used to established the simplified model and verify the analysis result. Some minor revisions are list below.

(1)  The type and the property definition of the Connector element in Section 2.2 are not clearly descripted by authors. And authors should explain why you chose the Connector element and what the difference between the Connector element and the Spring element.

(2)  The boundary condition and the loading method in Figure 3 are not mentioned in Section 2.2.

(3)  The hysteresis curves in Figure 8(b) are unclear between two models. This figure must be redrawn to let the readers distinguish the curves inside. And authors should explain the computational efficiency and accuracy between these two models.

(4)  What is the difference between the lateral bracing and the longitudinal bracing in Figure 9? However, the lateral bracing is only mentioned in Section 3. The missing description of the longitudinal bracing must be completed in Section 3.

(5)  Other major drawbacks of the paper are given in the following. The section geometry of the seismic support hangers in Figure 10 is completely missing and must be included.

(6)  The damping ratio of 0.2 is not accurate enough for elastic-plastic analysis in Line 315.

(7)  All the codes in Section 3.2 and Section 3.3 in the article are Chinese codes, but the reviewer and possibly most of the readers will not be familiar with Chinese standards. The author must describe the specific terms of these mentioned codes.

(8)  English language and style are minor spell check required. For example, the title of Section 2.1 should be called “General”. And the naming of the single-layer reticulated dome and seismic support hangers should be respectively consistent in this paper.

(9)  The format of reference is inconsistent.

Author Response

Dear reviewers,

We would like to thank the reviewers for providing us many constructive comments and suggestions. We have considered them carefully and have revised the manuscript accordingly.

Please see the attachment for detailed responses.

Reviewer 2 Report

The paper presents intriguing concepts and introduces novel observations. The submission holds promise for publication, but certain issues need to be addressed prior to its consideration. Here are some suggested revisions:

1. Check grammar.

2. Enhancements should be made to the introduction section. Explanation for the preference of arranging air ducts under a single-layer reticulated dome is needed. Additionally, comprehensive detailing of the fundamental components depicted in Figure 1 is essential.

3. References for the data presented in Table 2 should be cited and included.

4. The quality of figures requires improvement, particularly the clarity of experimental images in Figure 4 and Figure 5.

5. The suggested simplified mechanical model of the flange bolt joint in Figure 8 lacks clarity regarding how data is input for the connector element (manually or automatically). The discrepancy between the two hysteresis curves in Figure 8(b) also needs addressing.

6. Clarification regarding the damping ratio in elastic and elastic-plastic analyses at Line 315 is lacking. Detailed instructions on inputting damping data in the ABAQUS program are required.

The writting should be improved.

Author Response

(The authors gave the same response as above.)
